# Unique Features of Extremely Halophilic Microbiota Inhabiting Solar Saltworks Fields of Vietnam

**DOI:** 10.3390/microorganisms12101975

**Published:** 2024-09-29

**Authors:** Violetta La Cono, Gina La Spada, Francesco Smedile, Francesca Crisafi, Laura Marturano, Alfonso Modica, Huynh Hoang Nhu Khanh, Pham Duc Thinh, Cao Thi Thuy Hang, Elena A. Selivanova, Ninh Khắc Bản, Michail M. Yakimov

**Affiliations:** 1Institute of Polar Research, Institute of Polar Sciences, National Council of Research ISP-CNR, Via San Raineri 86, 98122 Messina, Italy; violetta.lacono@cnr.it (V.L.C.); gina.laspada@cnr.it (G.L.S.); francesco.smedile@cnr.it (F.S.); francesca.crisafi@cnr.it (F.C.); laura.marturano@isp.cnr.it (L.M.); 2Eni Rewind Environmental Engineering and Market Development/Servizi Laboratorio, EE&MD/SELAB, Contrada Cava Sorciaro 1, 96010 Priolo Gargallo, Italy; alfonso.modica@enirewind.com; 3Nha Trang Institute of Technology Research and Application, Vietnam Academy of Science and Technology, NITRA-VAST, Hung Vuong 2, Nha Trang 650000, Vietnam; khanhhuynh@nitra.vast.vn (H.H.N.K.); ducthinh@nitra.vast.vn (P.D.T.); caohang@nitra.vast.vn (C.T.T.H.); 4Institute for Cellular and Intracellular Symbiosis, Ural Branch, Russian Academy of Sciences, Pionerskaya Ul. 11, 460000 Orenburg, Russia; selivanova-81@mail.ru; 5Institute of Marine Biochemistry, Vietnam Academy of Science and Technology, IMBC-VAST, Hoang Quoc Viet 18, Nghia Do, Hanoi 100000, Vietnam

**Keywords:** extreme halophiles, solar saltern, microbiota, CPR bacteria, DPANN archaea

## Abstract

The artificial solar saltworks fields of Hon Khoi are important industrial and biodiversity resources in southern Vietnam. Most hypersaline environments in this area are characterized by saturated salinity, nearly neutral pH, intense ultraviolet radiation, elevated temperatures and fast desiccation processes. However, the extremely halophilic prokaryotic communities associated with these stressful environments remain uninvestigated. To fill this gap, a metabarcoding approach was conducted to characterize these communities by comparing them with solar salterns in northern Vietnam as well as with the Italian salterns of Motya and Trapani. Sequencing analyses revealed that the multiple reuses of crystallization ponds apparently create significant perturbations and structural instability in prokaryotic consortia. However, some interesting features were noticed when we examined the diversity of ultra-small prokaryotes belonging to *Patescibacteria* and DPANN *Archaea*. Surprisingly, we found at least five deeply branched clades, two from *Patescibacteria* and three from DPANN *Archaea*, which seem to be quite specific to the Hon Khoi saltworks field ecosystem and can be considered as a part of biogeographical connotation. Further studies are needed to characterize these uncultivated taxa, to isolate and cultivate them, which will allow us to elucidate their ecological role in these hypersaline habitats and to explore their biotechnological and biomedical potential.

## 1. Introduction

Solar salterns, renowned as extreme environments with salinities at or near saturation, are artificial thalassohaline–hypersaline ecosystems widely distributed worldwide. They capitalize on salinity gradients for solar salt production. The traditional method of solar salt-making involves concentrating brine and precipitating NaCl through natural evaporation in a series of interconnected ponds, each linked to the other through a common opening [1,2,3,4,5]. In addition to serving as excellent models for studying microbial diversity and ecology at various salt concentrations, marine solar salterns are valuable for investigating phenomena such as scarce water availability and desiccation, which are linked to ongoing global climate change [1]. Furthermore, they represent a valuable resource for exploiting extremophiles living at high saline concentrations, as they can be a source of unique enzymes with potential biotechnological applications [6,7,8]. Solar salterns sustain high microbial densities (10^7^–10^8^ cells/mL) due to the absence of predation and the presence of high nutrient concentrations [9]. Numerous studies have investigated the diversity of coastal solar salterns across various geographic regions, including Africa [10,11], Asia [12,13,14,15,16], Australia/Oceania [17], North and South America [18] and Europe [6,19,20,21,22,23]. The prokaryotic community of solar saltern brines has been the subject of study for decades, employing various approaches ranging from classical cultivation to culture-independent approaches, which use molecular techniques such as sequencing to directly analyze microbial DNA from environmental samples without the need for cultivation. The composition of microbial communities in salterns is considered stable over time [24,25] and is strongly influenced by salinity levels.

Culture-independent studies conducted on low-salinity samples (40–70 g/L) revealed a microbial community composition similar to that of a marine environment, with a predominance of bacterial phyla, including *Pseudomonadota*, *Cyanobacteriota*, *Bacillota*, *Bacteroidota*, *Actinomycetota* and *Mycoplasmatota*. At moderate salinity levels (130–180 g/L), bacterial phylotypes coexist with archaeal representatives of *Halobacteriota*. In contrast, high-salinity samples (>250 g/L) from solar salterns are characterized by a dominance of halophilic archaea, especially members of the families *Haloarculaceae*, *Halobacteriaceae* and *Haloferacaceae*, along with a minor contribution from bacterial phylotypes, mainly belonging to the *Bacteroidota* of the class *Rhodothermia*, family *Salinibacteraceae* [26,27,28]. Additionally, numerous studies have demonstrated that during the salt-making process, microbial communities, primarily consisted of haloarchaea, become trapped within halite a in salt-saturated environments, thereby avoiding the harsh, MgCl_2_-enriched bittern brine that remains after halite precipitates [29,30].

In this work, we studied extremely halophilic prokaryotic communities thriving in the salt crystallizer ponds of the Hon Khoi solar saltworks fields (HKsf), South Vietnam. This area is located in Khanh Hoa Province (14°34′23′′ N; 109°13′42′′ E), approximately 40 km from Nha Trang. Here, natural salt is harvested manually, without the use of heavy machinery, from shallow fields along Doc Let Beach. HKsf, covering an area of almost 400 hectares, is one of the largest salt-making areas in Vietnam, with production reaching almost 740,000 tons of salt per year, accounting for a third of Vietnam’s salt production. The salt harvesting season in the Hon Khoi salt fields occurs from January to July every year, with April to June considered the optimal time to obtain high-quality salt due to scorching temperatures during this period, accompanied by relatively low level of precipitation. Unlike the European method of collecting salt once or twice a year, in HKsf, the process is repeated many times. In short, seawater is directed from the East Sea into pre-drainage channels or basins, then into shallow pits (less than 50 cm depth), typically coated with HDPE liners, where the brine is left to evaporate for a very short period of time, usually no more than 10 days, after which the salt is manually extracted from these shallow pits, piled up and trucked away; the pits are then refilled with brine.

The diversity of extreme halophilic microbiota of HKsf salterns remains unexplored, thus potentially representing a hidden source of enzymes and secondary metabolites of high biotechnological demand. To fill this gap, an in-depth metabarcoding approach was conducted to study the prokaryotic diversity of crystallizer ponds and to characterize the bacterial and archaeal communities of HKsf salterns by comparing them with those of decommissioned solar salterns in North Vietnam. This was performed primarily to potentially identify geographically specific halophiles that could be linked to a location. Freshly collected halite samples were compared with brine and sediment samples for possible detection of geographically specific halophiles, regardless of the physical state of the collected samples. Finally, we integrated metabarcoding studies performed using freshly harvested halite and commercial salt samples obtained from Italian saltworks fields located in Trapani and Motya (western Sicily, Italy). This was performed to obtain molecular signatures that could be applied to track the salt samples using some geographical connotation of the product, linked to the production area.

To expand the possible existence of biogeographical regionalization of halophilic microbial communities thriving in Vietnamese saltworks sites, we also focused on an in-depth phylogenetic analysis of enigmatic prokaryotes belonging to the superphylum *Patescibacteria*, also called Candidate Phyla Radiation (CPR), and to the superphylum DPANN (this acronym comes from *Diapherotrites*, *Parvarchaeota*, *Aenigmarchaeota*, *Nanoarchaeota* and *Nanohaloarchaeota* candidate phyla). Members of these newly discovered taxa are typically ultra-small in size, with an average cell diameter of 200–450 nm, and possess highly reduced genomes (average size 1 Mbp) and minimal cellular activities, including limited metabolic potential [31,32,33]. As a consequence, these nano-sized organisms have adopted an obligate symbiotic or predatory lifestyle, relying entirely on their respective prokaryotic hosts [32,34,35,36,37,38]. They are extremely phylogenetically diverse, ubiquitous and abundant in microaerobic and/or anaerobic zones of freshwater lakes, sediments and groundwater [32,34]. There is very little evidence for the presence of both CPR and DPANN in hypersaline environments, and only members of the phylum *Nanohaloarchaeota* are commonly found worldwide in natural salt and alkaline lakes, marine solar brines and deep-sea hypersaline anoxic basins [14,25,39,40,41,42,43,44,45,46,47]. However, there is some evidence for the presence and even dominance of *Woesearchaeota* (now order *Woesearchaeales* in the phylum *Nanoarchaeota*) in halo-alkaline lakes and hypersaline sediments [45,48] and *Aenigmatarchaeota* (family *Haloaenigmarchaeaceae*) in slightly acidic hypersaline environments [49]. Recently, some *Nanohaloarchaeota* were stably cultivated in the laboratory as members of polysaccharide-degrading consortia [35,36]. Based on this finding, a similar enrichment approach was carried out with the Vietnamese samples to test whether some ultra-small prokaryotes could be an active ecophysiological component of extreme halophilic hydrolytic communities in the HKsf ecosystem.

## 2. Materials and Methods

### 2.1. Sampling

Two Vietnamese artisanal marine solar saltworks fields were chosen for sampling: the Hon Khoi Salt Fields, Nha Trang, Khanh Hoa Province (14°34′23′′ N; 109°13′42′′ E), one of the largest salt-making areas in Central Vietnam, and Hai Lý, Hai Hau, Nam Dinh Province (20°07′20.676′′ N; 106°17′45.959′′ E), a locally decommissioned salt field in northern Vietnam. A commercial salt sample was additionally collected from a fish sauce factory in Thành phố Cam Ranh city (Khanh Hoa Province), 60 km southeast of Nha Trang. For Italy, five samples of freshly harvested salt were collected from multiple artisanal solar saltworks in western Sicily: Trapani Salterns (37°58′49.90′′ N 12°29′42.00′′ E) and Motya Salterns (37°51′48.70′′ N, 12°29′02.74′′ E) (Figure 1). Sampling took place in 2022 (June and September) and 2023 (June). Brine temperature, salinity and pH were measured directly at the sampling site using a conventional thermometer, refractometer and pH meter (Table 1).

### 2.2. Geochemical Analyses and Salinity-Related Measurements

Correct quantification of anions and cations in hypersaline samples is a rather challenging analysis; in fact, the matrix shows high complexity due to both the variety of species in solution and the different concentration ranges in which they are present. Quantification of the major ions in this type of samples, such as chloride or sodium, is usually not a problem. However, the determination of other ionic species can be affected by lack of selectivity [50] and matrix effects [51]. Ion chromatography (IC) has been used to determine the anion composition of hypersaline samples using U.S. Environmental Protection Agency (EPA) Method 300.1 (1997) [52]. Ion chromatography with conductometric detection is the most popular technique for this determination due to its selectivity and sensitivity. The standard addition method has been used to compensate for matrix effects [51]. For quantification of cations, analyses were performed with Inductively Coupled Plasma- Atomic Emission Spectrometry (ICP-AES) using EPA Method 3005A (1992) [53] and E6010D 2014 [54].

#### 2.2.1. Anion Analysis

The ion chromatograph was a Dionex ICS-3000 (Thermo Fisher Scientific, Milan, Italy) equipped with an IP20 isocratic pump, a 25 µL loop, an IonPac AS9-HC separation column with an AG9-HC precolumn and a conductivity detector CD20 with a 4 mm ASRS Ultra II suppressor (Thermo Fisher Scientific, Milan, Italy). Chromeleon™ Chromatography Data System 7.3.2 CDS Software was used for instrument control, data acquisition and processing. The chromatographic and detection conditions were optimized to obtain an adequate resolution of peaks and sensitivity. The instrumental conditions were 9 mM Na_2_CO_3_, a flow rate of 1.0 mL/min and expected background conductivity of 24–30 µS. In the presence of a very high chloride concentration, the efficiency of early eluting peaks would be compromised due to the overloading effect. Pretreatment of the samples with Dionex OnGuard Ag followed by Dionex OnGuard H (Thermo Fisher Scientific, Milan, Italy) significantly reduced chloride and carbonate content, allowing accurate quantification. All chemicals used to prepare standard solutions were of analytical-reagent grade, namely, NaCl (≥99.8%) from Riedel-de Haën (Buchs, Switzerland), NaBr (≥99.7%) from VWR BDH Prolabo (Milan, Italy) and NaNO_3_ (≥99.0%) and Na_2_SO_4_ (≥99.0%) from Merck (Darmstadt, Germany). Na_2_CO_3_·10H_2_0 (≥99.0%) from Merck were used to prepare the eluent. All solutions were prepared with ultrapure water obtained from a Milli-Q Academic system (Millipore, Burlington, MA, USA). The impurities of the various salts did not affect the content of the other analytes significantly. Duplicate samples were also analyzed.

#### 2.2.2. Cation Analysis

Analysis of Na^+^, K^+^, Mg^2+^ and Ca^2+^ was performed using the Perkin Elmer OPTIMA 7300 DV ICP-AES apparatus (Perkin Elmer, Monza, Italy). Hypersaline samples contain more than 3 percent dissolved salts, which cause problems such as uneven sample transport rates and chemical and spectral interference. Samples were solubilized by acid digestion with a suitable mixture of HNO_3_ and HCl. The samples were then subjected to heating and reduced in volume. In the presence of particulate matter, after cooling, filtration was performed, and the samples were then brought to the final volume for subsequent analytical determinations using the mentioned ICP-AES technique. The analytical methods for both anion and cation determination were carried out by Eni Rewind Priolo Environmental Laboratory, which is accredited by the ACCREDIA/ILAC for the EPA 300.1 1997, EPA 3005 1992 and EPA 6010D 2018 methods, with accreditation number 0119L.

### 2.3. DNA Extraction and Purification

Brine samples (50–100 mL) were filtered through sterile Sterivex capsules (0.2 µm pore size, Merck Millipore, Burlington, MA, USA). A mass of 10 g of each Italian and Vietnamese salt sample was were weighed out and dissolved in prefiltered MilliQ water (final volume 40 mL); the obtained solution was filtered through 0.2 µm pore size membranes supported on a Swinnex filter holder with a 25 mm diameter (Merck Millipore, Germany). For each sample, filters were then used for DNA purification together with Sterivex capsules. Genomic DNA from sediments was extracted using the protocol of Hurt et al. [55]. Total DNA from both Sterivex and membrane filters was extracted using a GNOME DNA kit (MP biomedicals, Irvine, CA, USA). The extraction was carried out according to the manufacturer’s instructions. DNA was visualized on agarose gel (0.8% *w*/*v*). The quantity of DNA was estimated with a QubitTM 4 Fluorometer (Thermo Fisher Scientific, Waltham, MA, USA), metagenomic analysis was performed and *16S rRNA* gene amplicons were created.

### 2.4. Next-Generation Sequencing

For the sequencing of 16S rRNA gene amplicons, the V3–V4 hypervariable region of the 16S ribosomal RNA gene was sequenced using the primers previously described by Klindworth and coauthors [56]. DNA samples were PCR amplified and the amplicons were sequenced on the Illumina MiSeq platform by a commercial company (Macrogen Inc., Seoul, Republic of Korea) (http://dna.macrogen.com/main.do [accessed on 20 November 2023]). Library preparation and subsequent Illumina sequencing were performed according to standard protocols [57]. The sequences were stripped of barcodes and primers, and then sequences < 150 bp together with sequences with ambiguous base calls and those with homopolymer runs exceeding 6 bp were removed.

### 2.5. NGS Data Analysis

Reads were imported into the RStudio package (version 2023.06.0+421) [58] and analyzed using the DADA2 package (v 1.26.0) [59], as described in Crisafi et al. [60]. PCR biases were removed by applying a parametric error model constructed between the estimation of the error rate and the inference [59]. Chimeric sequences were identified and removed, and then the taxonomy was assigned using a DADA2 implementation of the naive Bayesian classifier method versus the Silva Database Release v.138.2 (https://www.arb-silva.de/documentation/release-138). The taxonomy table obtained (an ASV abundance table) was processed in R by means of the Phyloseq (v. 1.46.0), Vegan (v. 2.6.6.1) and Microbiome (v. 1.22.0) packages [61,62,63].

### 2.6. Statistical Analyses

A Phyloseq object was created from DADA2 results; possible contaminants and low-abundance ASVs (fewer than five reads) were removed. Diversity analyses were performed using the Phyloseq and Microbiome packages [61,62]. For each sample analyzed, alpha diversity was investigated in terms of the Shannon, Simpson and Chao1 indices. Beta diversity was calculated with UNIFRAC (weighted and unweighted distances) and the Jaccard and Bray–Curtis coefficients. The microeco R package (v. 1.8.0) was used to perform the Mantel test between the phylum abundance table and environmental variables [64].

### 2.7. Establishment of Polysaccharide-Degrading Enrichments and Isolation of an Axenic Xylanolytic Culture

The freshly harvested halite samples VCR and STP (5 g each) were used to establish polysaccharide-degrading enrichments. Based on knowledge of the successful isolation and maintenance of both chitino- and xylanolytic cultures [35,36], the same Laguna Chitin (LC) liquid mineral medium was used. For initial enrichments, 5 g of VCR halite was added to 70 mL of the LC medium, supplemented with either beechwood xylan (Megazyme (Wicklow, Ireland), catalogue number P-XYLNBE-10G) or pectin from citrus peel (Sigma Aldrich (St. Louis, MO, USA), catalogue number P9135). Both polysaccharides were sterilized separately by autoclaving and then added at a final concentration of 2 g/L to serve as growth substrates. The bacterium-specific antibiotics vancomycin and streptomycin were added (100 mg/L of each, final concentration) to prevent the growth of any halophilic bacteria. The same procedure was carried out for the STP halite sample, collected from an artisanal solar saltworks in Trapani. All four enrichments were incubated at 40 °C in tightly sealed 120-mL glass serum vials in the dark and without shaking for two to three months. The isolation strategy for axenic xylanolytic cultures consisted of several rounds of decimal-dilution transfers (each inoculation to fresh medium being a 10-fold dilution) to obtain active polysaccharidolytic enrichments, followed by a threefold repetition of serial dilution to extinction. The presence of *Nanohaloarchaeota* in the obtained enrichments was continuously monitored by PCR using the specially designed taxon-specific primers [35,36], and their 16S rDNA was cloned and sequenced.

## 3. Results and Discussion

### 3.1. Chemical Composition of Saltworks Fields

Among the nine sites sampled from Vietnam saltworks fields, physicochemical analyses were carried out at four geographically representative locations (Table 2). The striking differences between the three HKsf samples contrasted with the high similarity of hydrochemical parameters between the VS1 (HKsf) and Hai Lý crystallizer ponds. In both, an intermediate stage of transition from thalasso- to athalassohaline composition of brines was noted. On average, the predominance of major cations in these samples was in the order of Mg^2+^ > Na^+^ >> K^+^ >> Ca^2+^, compared to seawater’s proportions, Na^+^ >> Mg^2+^ > K^+^ ≥ Ca^2+^ (Table 2). Considering that these brines are nearly nine times more saline than seawater, these data are quite similar to those reported for the initial phase of late-stage evaporitic brines, formed during the evaporation path of seawater [65,66]. At this stage, halite begins to form, and this process is accommodated by a relative increase in the Mg^2+^ cation, which remains in the brine, in contrast to the precipitation of sodium. However, the VS8 brine was characterized by a strong enrichment of the Mg^2+^ cation (2.75 M), along with a noticeable drop in sodium in the brine by more than 4.4 times and 20%, compared to the VS1 and Hai Lý brines and the initial seawater, respectively. Thus, the VS8 brine at the time of sampling represented the final phase of the seawater evaporation path, in which only a minor quantity of sodium was still present. The hydrochemical characteristics of the VS7 brine, less saline than the other brines analyzed, were more interesting. In fact, despite being 4.3 times saltier than seawater, it contained only 1.5 times more sodium, while the content of Cl^−^ and especially Mg^2+^ and Br^−^ was 4, 17 and 14 times higher, respectively. The relatively high concentrations of Mg^2+^ and Br- are very indicative of the origin of the VS7 brine, since these concentrated ions are reported to resist crystallization during seawater evaporation and they are not predominantly incorporated into most of the precipitates [66,67,68]. Considering Br^−^ as a model ion for comparison with the major ions Na^+^ and Cl^−^ in brines, we noticed that VS7 deviated significantly from the seawater evaporation path in both plots, which was not the case with the Mg^2+^–Br^−^ correlation, which fully followed this path (Figure 2). This unusual hydrochemistry is easily explained by the abovementioned peculiarity of the salt harvesting process in Vietnam, namely, the frequent reuse of crystallizer ponds throughout the year. In other words, after the removal of the precipitated halite, the bitter brine seemingly enriched in Mg^2+^ and Br^−^, which remained in the VS7 pool in unknown quantities, was diluted with seawater, slightly evaporated in pre-drainage saline saltern pond(s), to continue the salt extraction process.

### 3.2. Alpha and Beta Diversity

To investigate the composition of the prokaryotic communities in different solar salterns and freshy harvested salt samples, representative unique amplicon sequence variants (ASVs) were analyzed and compared to the microbial reference database for identification. After filtering, a total of 747,857 16S rRNA sequences were obtained, ranging from 4028 to 61,118 sequences per sample, which belonged to a total of 4900 distinct ASVs (Table 3). The Simpson and Shannon diversity indices, which take into account both species richness and species evenness, ranged from 12.12 to 320.05 and from 2.99 to 6.0, respectively. The highest value was obtained for sediment sample VS9, indicating that it had the highest diversity within its community. The lowest value was found in the freshly collected salt sample CP, which correspondingly exhibits the lowest diversity.

Beta diversity was measured as weighted and unweighted UniFrac and plotted using PCoA (principal coordinates analysis) (Appendix A). PCoA of microbial community data was used as valuable tool for exploring and visualizing the complex relationships between Italian and Vietnamese microbial communities. Considering the varying abundances of different ASVs detected in each sample, the PCoA plot generated by the weighted UniFrac distance matrix shows that neither sample matrices nor salinity nor provenance is a significant factor influencing the differences in microbial community composition between samples. The unweighted UniFrac distance, which only takes into account the presence and/or absence of different ASVs along with their phylogenetic distance, better reflects the similarities between samples. The freshly harvested halite samples were clustered together and far from the others. The stress value of 0.11 in the NMDS analysis using the Jaccard distance as the similarity metric can be considered acceptable and suggests that the NMDS plot provides a good representation of the relationships between samples in the microbial community compositions (Appendix A). This was confirmed by Bray–Curtis analysis, which indicates that the spatial configuration obtained by the NMDS analysis closely approximates the similarities/dissimilarities in the sample data.

### 3.3. General Microbiome Profiling

The distribution of bacterial and archaeal taxonomic profiles is shown in Figure 3. Within the samples, 280 bacterial genera belonging to thirty-four recognized phyla and 49 archaeal genera from five recognized phyla (GTDB taxonomy; https://gtdb.ecogenomic.org/ [accessed on 20 June 2024]) in total were identified. Notably, archaeal abundance ranged from 80% to 90% in freshly harvested VFS halite and VS6 brine collected from Hon Khoi Salt Fields, as well as in commercialized salt VCR from Cam Ranh and Hai Lý brine (Figure 3A). Sample CP, consisting of freshly harvested halite from Motya (Italy), exhibited a similar predominance of archaea (91%). However, in other samples, which also included freshly collected halite samples and brines with salinities exceeding 250 g/L, archaeal presence ranged between 20% and 60%. This deviation was in some contrast to the results of previous studies [12,13,14,15,16,17,18,19,20,21,22,23,24,25,26,27,28,69,70] and may be attributed to the aforementioned distinct characteristics of Vietnamese solar salterns, the numerous reuses of the crystallizer ponds with the addition of slightly evaporated seawater (50–100 g/L) and the composition of the analyzed samples, which comprised a mixture of salts, sediments, microbial mats and brines. The core tidyverse packages [71] and the phangorn [72], dendextend [73], vegan [63], ggdendro [74], ggsci [75] and cowplot [76] libraries were used to create bar charts with hierarchical clustering (Figure 3).

As we mentioned above, alpha and beta diversity data did not reveal a clear phylogenetic relationship or grouping of microbial populations by source location (Appendix A). Additional statistical analysis was performed to monitor the environmental settings that are likely to be involved in the shaping of bacterial populations. Namely, the Mantel test, implemented using the microeco R package [64], was applied to evaluate the correlation between the main physicochemical parameters and the structure of the recognized microbial phyla found in all the analyzed samples. Several important factors with significant *p* and *r* values, such as salinity, pH, temperature and oxygen availability (measured in situ as redox potential), emerged as determinants of the presence or even abundance of a certain phylum (Figure 4). The test was based on the Bray–Curtis ASV dissimilarity matrix. The *r* value represents the correlation between the environmental factors and the ASV matrix, while *p* determines whether the correlation coefficient *r* is statistically significant. The most statistically significant positive correlations were observed between salinity (S) and dominance of halophilic archaea belonging to the phyla *Halobacterota* and *Nanohaloarchaeota* (Figure 4, Table 4), as indicated by high *r* values (0.899 and 0.663, respectively) and highly significant *p* values (0.001 and 0.002, respectively); the low false discovery rate (FDR) (0.013) suggested the high reliability of these results. These findings suggest a direct relationship between increasing salinity and the abundance of these extremely halophilic archaeal groups, which is consistent with their physiology and inability to function under low-salinity conditions. Strong positive correlations were also observed between pH and the dominance of the phyla *Bacillota* and *Bipolaricaulota* (*p* < 0.01); an FDR value of 0.020 indicated moderate correlations that are still significant (Figure 4, Table 4). Their respective *r* correlation coefficients of 0.503 and 0.852 indicate a substantial relationship between pH and the abundance of these bacterial groups. The same trend, although to a lesser extent, was observed between oxygen concentration, measured as reduction–oxidation potential, and the emergence of the phyla *Bacillota*, *Bipolaricaulota*, *Fusobacterota* and *Patescibacteria*. This observation is consistent with the fact that many representatives of these phyla prefer to thrive in microoxic and/or anaerobic environments.

### 3.4. Bacterial Diversity of Brine, Sediment and Salt (Halite) Samples

*Bacteroidota* and *Pseudomonadota* were the dominant bacterial phyla consistently distributed across the salinity gradient and omnipresent in both Vietnamese and Italian samples during salt production (Figure 3B). However, these phyla exhibited marked differences in abundance across sites. *Bacteroidota* displayed a wide range, from a low of 8% at Hai Lý HL350 to a high of 92.9% at the hypersaline VS8 site (late-stage brine). Conversely, *Pseudomonadota* demonstrated an opposite pattern, with a maximum abundance of 91.3% at Hai Lý HL350 and a minimum of 2.6% at VS8. *Bacillota* was also a consistently detected phylum, though less abundant than *Bacteroidota* and *Pseudomonadota*, and was in most samples except the fresh salt from HKfs. *Desulfobacterota* represented a smaller proportion of the community, ranging from 5 to 14.1% in anaerobic HKfs samples, with a peak abundance of 24.4% in the Italian salt sample SM19 (Figure 3B). At the genus level (Figure 5), ASVs belonging to the phylum *Pseudomonadota* were primarily classified as *Gammaproteobacteria*, representing over 50% of the bacterial community in both Hai Lý samples (86.2% and 61.2% in HL450 and HL345, respectively) and in the Vietnamese salt samples VS3 (52.9%), VFS (70%) and VCR (52.2%). *Alphaproteobacteria*, on the other hand, never exceeded 30% of the bacterial population, with a peak abundance of 28.8% in the VS4 (brine and salt) sample. Delving deeper into the genus level, members of the genus *Salinibacter* of the phylum *Bacteroidota* (former phylum *Rhodothermota*) consistently appeared in all analyzed samples excluding HKsf’s less saline VS2, VS5 and VS7 brines. The highest relative abundance of *Salinibacter* was observed in the Italian and Vietnamese freshly harvested halite STP and VS8 brine (69.2% and 91.3%, respectively). It is well-know that *Salinibacter* uses the “salt-in” strategy known from the archaeal *Halobacteriota* and is a major component of hypersaline aquatic ecosystems worldwide ([77] for further references). In contrast, to the ubiquitous *Salinibacter*, the gammaproteobacterium *Halovibrio* was exclusive to brine and predominated in the Hai Lý samples, comprising 45.8% and 38.75% of the community, respectively. Overall, brine samples exhibited a higher diversity than halite, which was indicated through the presence of sequences belonging to a higher number of phyla and a correspondingly greater number of bacterial genera.

### 3.5. Archaeal Diversity of Brine, Sediment and Salt (Halite) Samples

Among *Archaea*, *Halobacterota* was the dominant archaeal phylum in all but one of the tested samples (brines, microbial mats and salts), accounting for up to 99% of the total archaeal communities in Hai Lý HL345 brine (Figure 3C). In the VS7 brine, however, they represented only 14% of all archaea, likely due to the unusual hydrochemistry of this site as mentioned above. Apparently, prior to our sampling, the precipitated halite had just been removed from the pool along with most of haloarchaeal cells embedded in the salt crystals, thus artificially enriching the remaining bitter brine with *Nanohaloarchaeota*, which ultimately comprised 78.5% of the archaeal population. The members of *Aenigmarchaeota* made up the remaining 7.5% of archaea in VS7 brine, and their presence in this brine is a likely the result of recent addition of slightly evaporated seawater from the saltworks’ pre-drainage channels to repeat the salt harvesting process. Indeed, these typically marine archaea, first discovered and named as the “Deep Sea Euryarchaeotic Group” (DSEG) [78], disappeared with increasing total salinity in the other, more saline brine samples. Representatives of two other archaeal phyla, *Thermoplasmatota* and *Nanoarchaeota* (former *Woeasearchaeota*), showed the same tendency towards salinity intolerance and were found only in the less saline ponds VS2 and VS5, albeit in significant numbers (relative abundance 12–16%).

At the genus level, for *Halobacterota*, we found a typical trend of increasing haloarchaeal diversity associated with increasing brine salinity [6,26,27,28,79,80], with up to 20 haloarchaeal genera detected in the saltiest brines (Figure 6). This is almost three times the number of haloarchaeal genera found in less saline samples (six genera on average). In contrast to other studies conducted at European saltworks [6,27,28,30], which traditionally harvest halite once or twice a year, the multiple reusage of crystallization ponds in Vietnam apparently creates significant perturbations and structural instability in consortia. In fact, when examining the haloarchaeal community across all brine and halite samples, we did not detect any minimally defined structure at the genus level. They were all quite diverse, even those in close proximity. However, some interesting features were noticed, at least two of which deserves special attention. First, the genus *Halorubrum* was found among the most represented taxa across the investigated brines and salts from solar salterns in both Vietnam and Italy (present in 17 of all 19 analyzed samples), reaching its highest relative abundances in freshly collected halite from CP (Italy) and VFS (Vietnam) with relative abundances of 90% and 57% of archaeal sequences, respectively (Figure 6). Studies examining commercial salt derived from marine solar salterns [81,82,83,84] and ancient halite [85,86,87,88] have demonstrated that haloarchaea that dominate brine communities are often present with reduced abundance or undetectable in halite communities. This does not appear to be the case for *Halorubrum*, as its dominance (up to 51% of relative abundance) was also detected in many of HKsf and Hai Lý brines (Figure 6). These data correspond well with the dominance of *Halorubrum* in the brines of solar salterns and freshly harvested halite of Trapani, where its mean relative abundance was found to be 55.2% [30]. Secondly, the highest relative abundance of the genus *Halobaculum* was found only in the Hon Khoi salt field samples (up to 10% in brines and up to 16% in halite samples, respectively), while their presence as a minority or even their absence was detected in all other analyzed samples, both Vietnamese and Italian. Though a study of *Halobacteria* associated with halite crystals collected from coastal salterns of Western Europe, the Mediterranean and East Africa, yielded little support for the existence of biogeographical patterns for this group of *Archaea* [77,83], the presence of the genus *Halobaculum* among the most represented taxa in the HKsf samples can be taken into consideration as a strong geographical connotation that is quite indisputably linked to the Hon Khoi salt production area.

### 3.6. Diversity of Nano-Sized Microbiota

To further explore the possible existence of biogeographical regionalization of halophilic microbial communities thriving in Vietnamese saltworks sites, we conducted an in-depth phylogenetic analysis of ultra-small prokaryotes belonging to DPANN *Archaea* and CPR *Bacteria* (Figure 7). Representatives of DPANN *Woesearchaeota* (now order *Woesearchaeales* in the phylum *Nanoarchaeota*) taxon were found in all HKsf brines and halite, reaching the 16.4% (VS5 brine) and 6.5% (VS3 halite) of all archaeal community, respectively. Most of the ASVs associated with HKsf *Woesearchaeota* formed a deeply branched cluster, DBCVW#1, together with a single similar riboclone, Y9A (EF106719) (98.53% of identity), recovered from the Guerrero Negro Baja endoevaporitic crust community [89]. This appears to be a very specific halophilic cluster of *Woesearchaeales*, as we could not find any other relative sequences in the NCBI database with more than 88% identity.

The phylum *Nanohaloarchaeota* was, as expected, abundant in all analyzed samples, reaching a peak relative abundance of 78.5% in the brine sample VS7 (Figure 3C), likely for the reasons discussed above. Screening for the HKsf-specific signatures revealed the presence of at least two deeply branched clusters, found only in Hon Khoi brine and halite samples (Figure 7). The first cluster, DBCVN#1, consisting of 14 distinct ASVs, was most similar (93.7–97.2% identity) to cultured nanohaloarchaeon *Candidatus* Nanohalococcus occultus [36,90] with no other related sequences in the NCBI database possessing greater than 93% identity (it is worth to notice that all riboclones similar at this level were recovered exclusively from Asian hypersaline lakes). The second cluster, DBCVN#2, consisting of 12 distinct ASVs, was most similar (97.98% identity) to four riboclones AB576-D04, AB577-G21, AB577-N13 and AB578-J17, all recovered from Santa Pola salterns (Spain) [91]. The next most similar nanohaloarchaeon, with 95.1% identity, was cultured *Candidatus* Nanohalobium constans [35]. Both these cultured nanohaloarchaea are characterized by their interspecies interactions with polysaccharidolytic haloarchaea, which can be assigned to the classical types of either mutualistic symbiosis or pure commensalism. *Candidatus* Nanohalobium is associated with a chitin-degrading *Halomicrobium* strain, while *Candidatus* Nanohalococcus occultus were grown in a xylan-degrading trinary culture with *Haloferax lucentense* and *Halorhabdus* sp. (*Halobacteriota*). These data indicate that the ectosymbiotic nanohaloarchaea found in such overwhelming abundance in the HKsf ecosystem are likely an active ecophysiological component of extreme halophilic communities that can degrade polysaccharides in hypersaline environments.

Another notable feature of the Hon Khoi saltworks ecosystem was the presence, sometimes even in significant numbers (representing 12.37% of the bacterial community in the VS7 brine), of organisms belonging to the superphylum *Patescibacteria*, also called Candidate Phyla Radiation (CPR) (Figure 8). The phylogenetic affiliation of 64 related ASVs showed a distribution in at least 11 different classes or clades of *Patescibacteria* (Figure 8). Among all 5687 CPR-related reads retrieved, nearly two-thirds were from the less saline VS7 brine (66.7%). Their presence gradually decreased with increasing salinity, from 10.6% in the VS5 brine to 4.6% in the VS4 brine (204 g/L and 425 g/L, respectively). None of CPR-related sequences were recovered from either the Hai Lý brines or the HKsf and VCR halite samples. Interestingly, only CPR members distantly related to the cluster of *Candidatus* Uhr- and Kerfeldbacteria were detected in a single Italian halite sample, SSR (8.7% of all CPR-related sequences). In-depth phylogenetic analysis revealed no clear grouping with the CPR riboclones from the NCBI database based on the isolation source, the ecosystem type/trophic state or the presence/absence of oxygen. Many of the related NCBI sequences were from very different ecosystems, from soil, marine sediments, mud volcano, fresh water lakes and hypersaline ecosystems (Figure 8).

Similar to *Nanohaloarchaeota*, two deeply branched clusters of CPR bacteria were found to be highly specific to the Hon Khoi saltworks field ecosystem. The first cluster, DBCVP#1, was detected only in VS2 and VS7 brine samples. It consists of 11 distinct ASVs and, together with five uncultured riboclones (NK2_65, ELA_111314_OTU_5299, SEAB1DG111, SEAB1BH081 and SEAB1BC021) recovered from wetland soils and freshwater lakes, was distantly related to *Candidatus* Pacebacteria (currently the class *Microgenomatia*) with no other relatives in the NCBI database with over 91% identity. Much more intriguing was the discovery of a second, class-level cluster, DBCVP#2, found only in the VS1, VS4 and VS7 brines and grouping 17 individual ASVs, unifying 2473 sequences taxonomically assigned to *Patescibacteria*. Members of this cluster accounted for 100%, 17.1% and 62.8% in brines VS1, VS4 and VS7, respectively. Cluster DBCVP#2 appeared to be halophilic, as all members were very distantly related (90.2–94.6% identity) only to two bacterial riboclones, OM_int_bact547 and LGNSa2_BC_059, both retrieved from hypersaline ecosystems [92,93]. We could not find any other relatives in the NCBI database with the sequence identities greater than 85%.

Because most *Patescibacteria* remain uncultured, it is difficult to make any assumptions about their specific hosts in the Hon Khoi saltworks field ecosystem. However, some halophilic representatives of the class *Candidatus* Gracilibacteria have been maintained in stable binary cultures with gammaproteobacterial hosts [37], abundant in HKsf brines. Thus, it is likely that *Patescibacteria* are important components of the HKsf microbiome and may harbor unique metabolic traits that enable them to thrive under hypersaline conditions. This finding highlights the complexity of food webs in the Hon Khoi saltworks field ecosystem, with symbiotic and predatory nutritional strategies existing there.

## 4. Conclusions

In this study, we describe the first detailed data on halophilic prokaryotic communities from different crystallizer ponds and halite samples from the Hon Khoi and Hai Lý saltworks fields (Vietnam), as well as halite samples collected from several artisanal solar saltworks in western Sicily (Italy). Among *Archaea*, the extremely halophilic phyla *Halobacterota* and *Nanohaloarchaota* appeared as the dominant population in all analyzed samples (brines, microbial mats and salts), accounting for up to 99% of the total archaeal communities in brine HL345. Noteworthy, the highest relative abundance of the genus *Halobaculum* was found specifically in the Hon Khoi salt field samples, which can be considered a biogeographical connotation.

In the same vein of the likely existence of other biogeographic patterns, we examined the diversity of the ultra-small prokaryotes belonging to *Patescibacteria* (CPR) and DPANN *Archaea*. Surprisingly, we found at least five deeply branched clades, both bacterial and archaeal, that appeared to be highly specific to the Hon Khoi saltworks field ecosystem. Out of the two CPR clusters, DBCVP#2, consisting of 17 distinct ASVs, together with only two similar sequences (90.2–94.6% identity) known, retrieved from hypersaline ecosystems of Oman and Mexico, appears represent a novel halophilic class-level taxon within the phylum *Patescibacteria*. The same is true for most of the ASVs associated with HKsf *Woesearchaeales* (phylum *Nanoarchaeota*), as they formed a deeply branched halophilic cluster DBCVW#1, along with a single similar riboclone recovered from the same hypersaline ecosystem of Mexico as the CPR-associated clone mentioned above. Since most of ultra-small prokaryotes remain uncultured, it is difficult to make assumptions about their specific hosts, their roles and functional activities in the Hon Khoi saltworks field ecosystem. Only two nanohaloarchaeal clades, DBCVN#1 and DBCVN#2, were found be distantly related to the cultivated *Candidatus* Nanohalobium and *Candidatus* Nanohalococcus, both of which are members of polysaccharide-degrading consortia. This finding may suggest that ectosymbiotic ultra-small prokaryotes or at least nanohaloarchaea are an active ecophysiological component of extreme halophilic hydrolytic communities in the HKsf ecosystem.

The discovery of previously overlooked groups of halophilic ultra-small CPR and DPANN organisms expands our knowledge of their econiche spectra and suggests their active participation in biogeochemical cycling and functioning in hypersaline ecosystems, which may also imply control over the eubacterial and haloarchaeal populations that bloom there. Further studies are needed to characterize these uncultivated taxa, as well as attempts to isolate and cultivate them, which will allow us to elucidate their ecological role in these hypersaline habitats and explore their biotechnological and biomedical potential. Finally, this study helps fill the gaps in our understanding of the prokaryotic diversity and resources of Vietnamese saltworks.

## Figures and Tables

**Figure 1 microorganisms-12-01975-f001:**
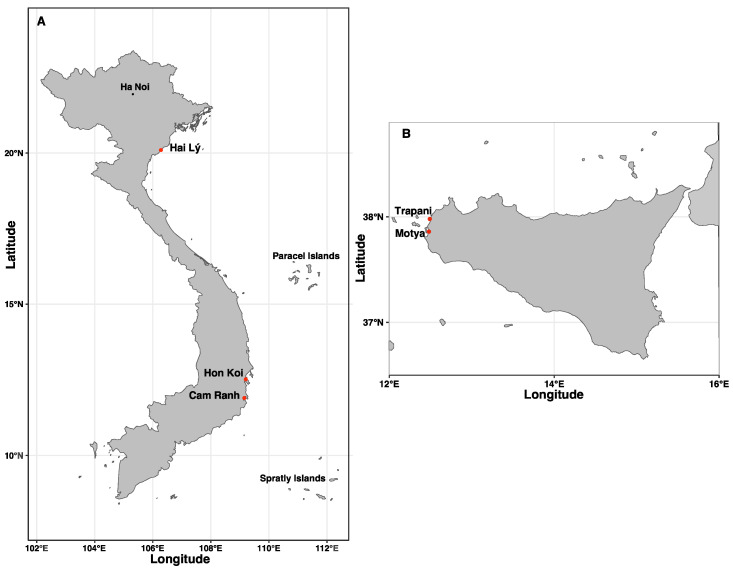
Locations of sampling sites in Vietnam (**A**) and Italy (**B**).

**Figure 2 microorganisms-12-01975-f002:**
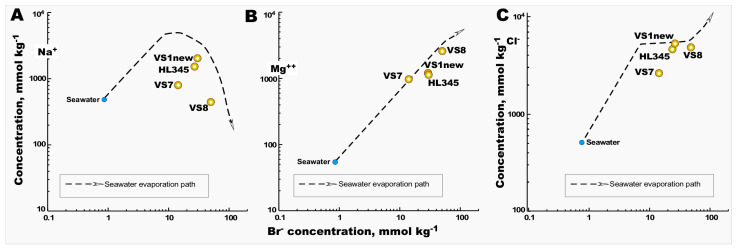
Evaporation path of seawater in the Na^+^ vs. Br^−^ (**A**); Mg^2+^ vs. Br^−^ (**B**) and Cl^−^ vs. Br^−^ plots (**C**). Average brine compositions of Vietnamese samples are indicated as yellow bubbles. Seawater values are shown as blue bubbles.

**Figure 3 microorganisms-12-01975-f003:**
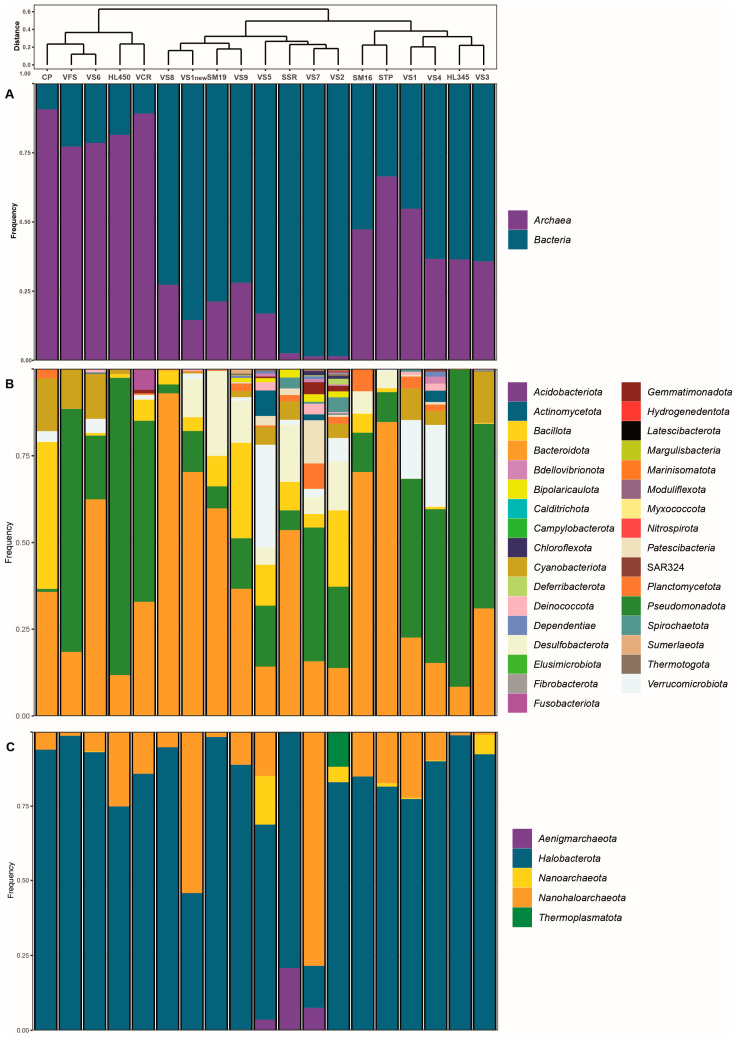
Bar charts of *Bacteria* and *Archaea* identified in all analyzed solar salterns and salt (halite) samples (**A**) and their relative sequence abundance at the phylum level: (**B**) *Bacteria*; (**C**) *Archaea*. The hierarchical clustering based on the Bray–Curtis dissimilarity matrix of community compositions is shown above the bar charts.

**Figure 4 microorganisms-12-01975-f004:**
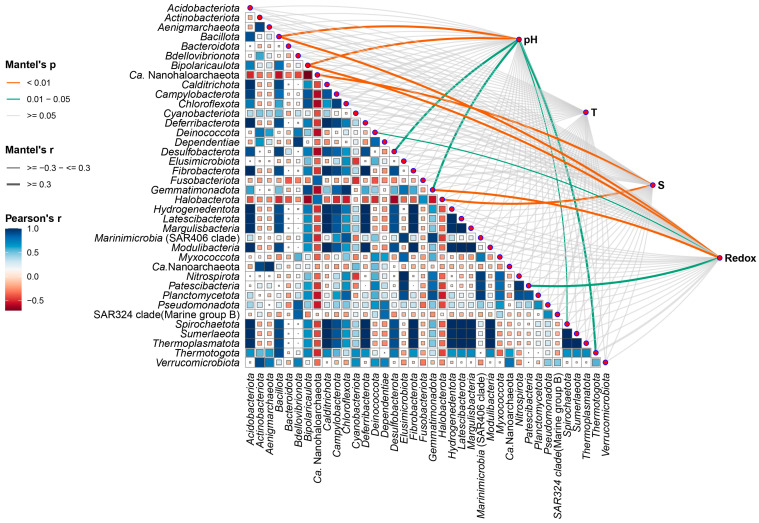
The Mantel test demonstrates the correlation between targeted phyla, present at relative abundances > 1.0%, and physicochemical factors. Mantel’s *r* coefficient quantifies this relationship, with line width representing correlation strength and color indicating statistical significance based on 9999 permutations (*p* < 0.01, *p* < 0.05). A Pearson correlation coefficient matrix reveals the interrelationships among dependent variables.

**Figure 5 microorganisms-12-01975-f005:**
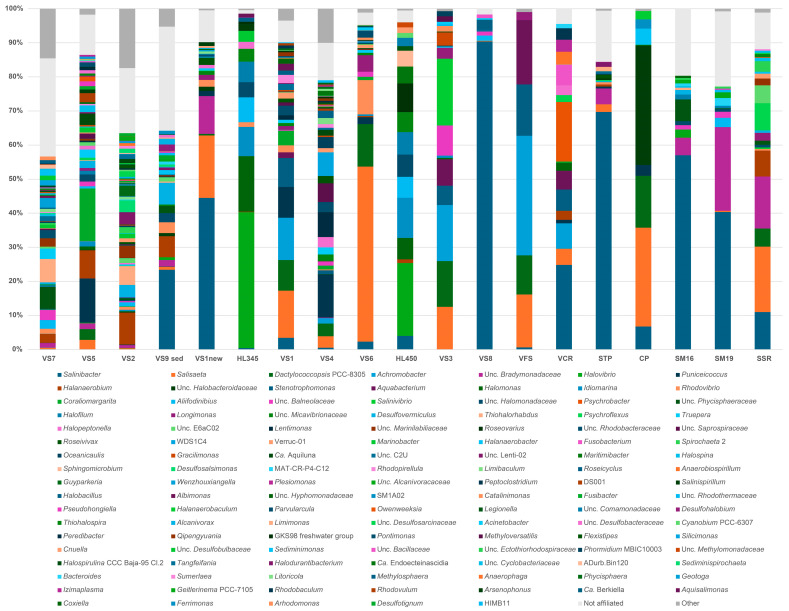
Bar charts of taxonomic classification (at genus level) of *Bacteria* identified in all analyzed solar salterns and salt (halite) samples. Only genera whose average relative abundances (>1.0%), based on 16S rRNA gene analysis, are shown. Genera with ambiguous affiliation were combined and are shown as NA. Genera with relative abundances less than 1.0% were combined and are depicted as Other. A complete list of all identified genera and their relative abundances is reported in Appendix A.

**Figure 6 microorganisms-12-01975-f006:**
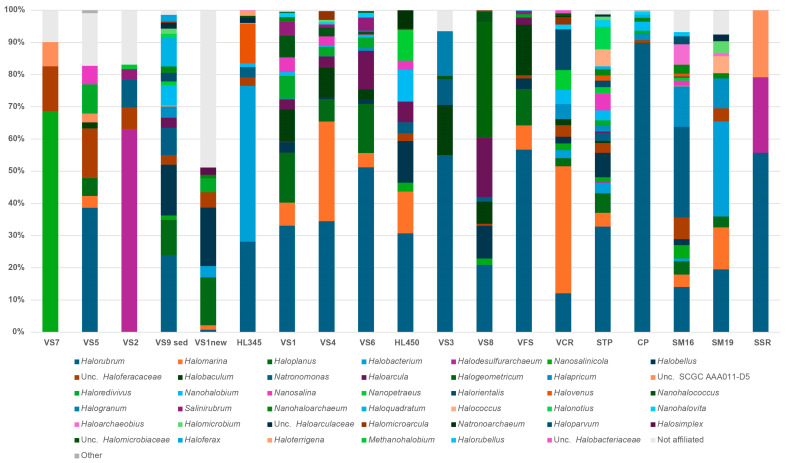
Bar charts of taxonomic classification (at genus level) of *Archaea* identified in all analyzed solar salterns and salt (halite) samples. Only genera whose average relative abundances (>1.0%), based on 16S rRNA gene analysis, are shown. Genera with ambiguous affiliation were joined and shown as Not affiliated. Genera with relative abundances less than 1.0% were joined and depicted as Other. A complete list of all identified genera and their relative abundances is reported in Appendix A.

**Figure 7 microorganisms-12-01975-f007:**
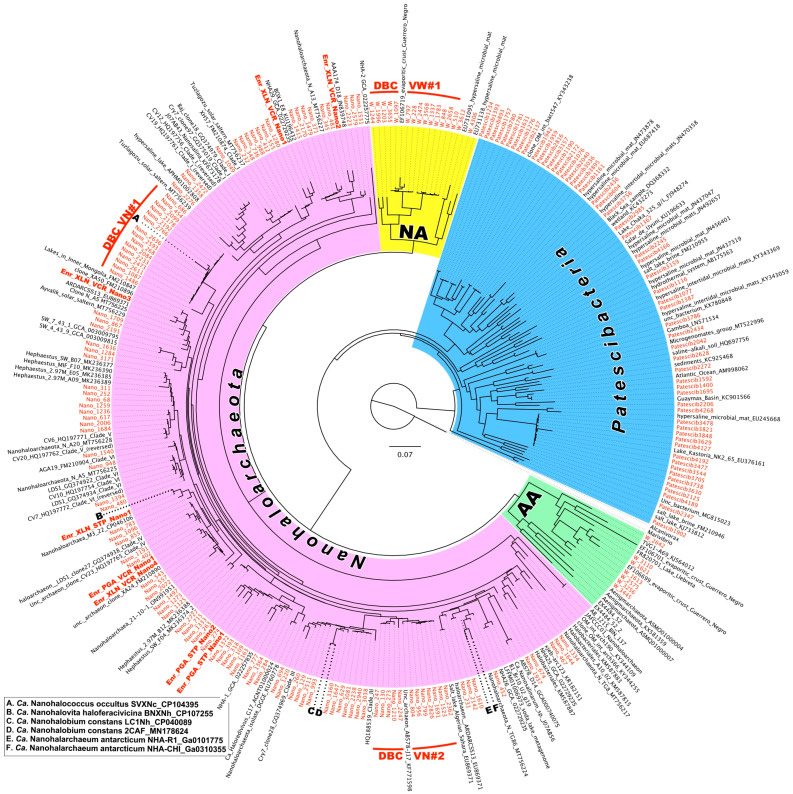
Randomized Axelerated Maximum Likelihood (RAxML) tree of 16S rRNA genes of the superphylum *Patescibacteria* and different phyla of the superphylum DPANN Archaea. A phylogeny was generated using 110 (*Nanohaloarchaeota*), 16 (*Nanoarchaeota*), 7 (*Aenigmatarchaeota*) and 62 (*Patescibacteria*) distinct ASVs obtained during this study (highlighted in red) and reference GenBank riboclones. The tree was constructed based on taxonomy, assigned to ASVs using a naïve Bayesian classifier method against the Silva Database v138 (https://www.arb-silva.de/documentation/release-138 and https://zenodo.org/record/4587955#.YgKJlb_MJH4 [both last accessed on 20 June 2024]). After alignment, the neighbor-joining algorithm of the ARB v.7.0 software package was used to generate the phylogenetic trees based on distance analysis for 16S rRNA. The tree was additionally inferred in the maximum likelihood framework using the MEGA v.6.0 software. The robustness of inferred topologies was tested by bootstrap resampling using the same distance model (1000 replicates of the original dataset). The scale bar represents the average number of substitutions per site. Deeply branched clades and nanohaloarchaeal sequences obtained in polysaccharidolytic enrichments are highlighted in bold red color. Cultivated nanohaloarchaea (A–F) are shown in insert. Abbreviation used: AA, *Aengimatarchaeota*; DBCVN, deeply branched clade of Vietnamese *Nanohaloarchaeota*; DBCVW, deeply branched clade of Vietnamese *Woesehaloarchaeota*; NA, *Nanoarchaeota*.

**Figure 8 microorganisms-12-01975-f008:**
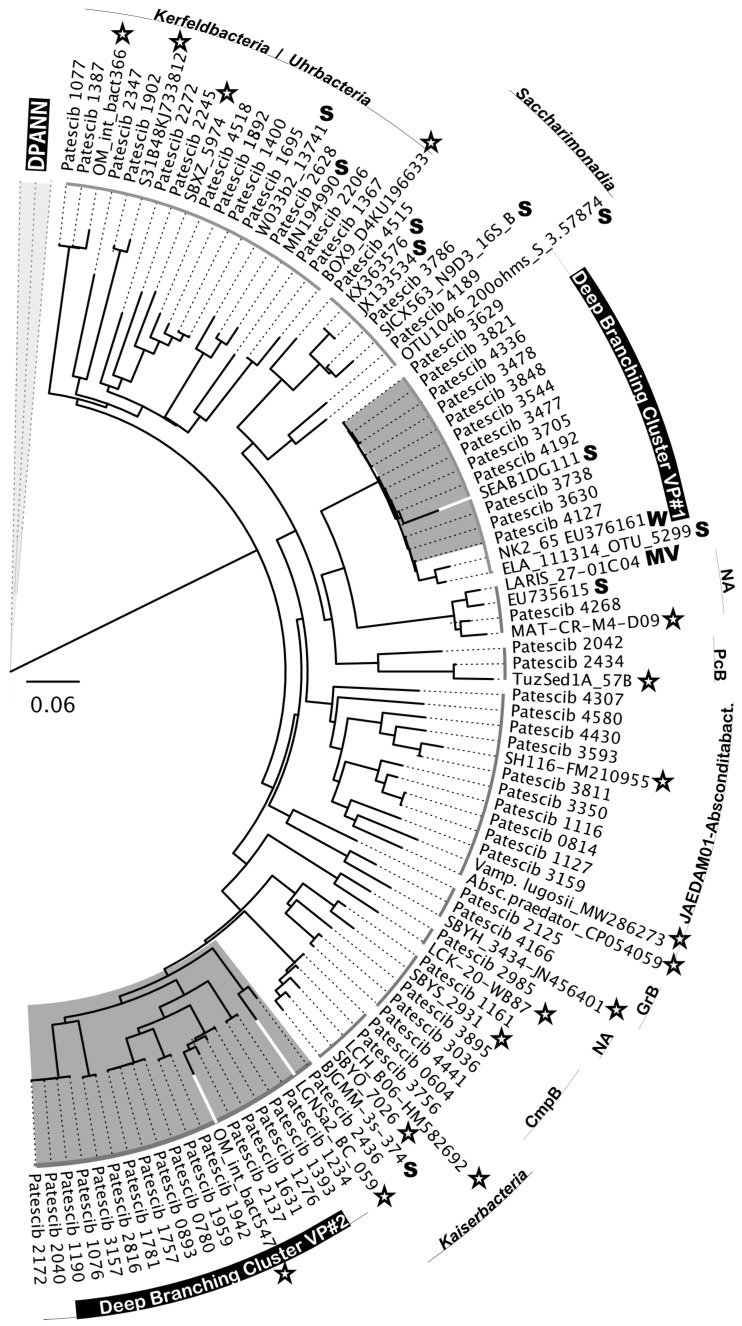
Randomized Axelerated Maximum Likelihood (RAxML) tree of 16S rRNA genes of different classes of the superphylum *Patescibacteria*. A phylogeny was generated using 64 distinct ASVs and reference GenBank riboclones. The robustness of inferred topologies was tested by the bootstrap resampling using the same distance model (1000 replicates of the original dataset). Scale bar represents the average number of substitutions per site. Reference GenBank riboclones obtained from hypersaline environments, soil, groundwater and mud volcanoes are designated by stars and the letters S, W and MV, respectively. Abbreviation used: Absconditabact., *Candidatus* Absconditabacteria; CmpB, *Candidatus* Campbellbacteria; GrB, *Candidatus* Gracilibacteria; NA, not affiliated; PcB, *Candidatus* Pacebacteria.

**Table 1 microorganisms-12-01975-t001:** Sample properties and sampling site descriptions.

	Site Name	Label	Latitude	Longitude	Salinity (g/L) *	pH	mV	T (°C)	Sample Type	Date
VIETNAM	HKsf	VS1	12°32′7.732′′ N	109°12′49.244′′ E	375	7.07	21	40.8	Salt and brine	6 June 2022
	HKsf	VS2	12°32′6.194′′ N	109°12′49.752′′ E	220	7.75	−62	40.8	Microbial mat and brine	6 June 2022
	HKsf	VS3	12°32′6.194′′ N	109°12′49.752′′ E	1000	NA **	NA	NA	Salt	6 June 2022
	HKsf	VS4	12°31′42.2′′ N	109°13′23.167′′ E	425	6.86	2.4	44.9	Brine and salt	6 June 2022
	HKsf	VS5	12°32′5.464′′ N	109°12′51.347′′ E	204	7.75	−53	38	Microbial mat and brine	6 June 2022
	HKsf	VS6	12°32′5.464′′ N	109°12′51.347′′ E	450	6.86	29	37.8	Microbial mat and brine	6 June 2022
	HKsf	VS7	12°32′16.735′′ N	109°12′26.827′′ E	165	7.6	−50	36.5	Brine and sediment	15 June 2023
	HKsf	VS1new	12°32′16.85′′ N	109°12′26.553′′ E	340	6.9	12	44	Brine	15 June 2023
	HKsf	VS8	12°32′20.868′′ N	109°12′31.428′′ E	320	6.9	32	42.8	Brine	15 June 2023
	HKsf	VS9sed	12°32′23.496′′ N	109°12′31.464′′ E	340	6.8	−243	60.2	Brine, microbial mat and sediment	15 June 2023
	HKsf	VFS	12°32′0.164′′ N	109°12′49.742′′ E	1000	NA	NA	NA	Salt	6 June 2022
	Cam Ranh	VCR	11°55′17.18′′ N	109°09′32.870′′ E	1000	NA	NA	NA	Salt	7 June 2022
	Hai Lý	HL450	20°07′20.28′′ N	106°17′45.962′′ E	450	8.5	3.9	42	Salt and brine	19 June 2023
	Hai Lý	HL345	20°07′20.64′′ N	106°17′45.961′′ E	345	8.0	8.7	40.8	Salt and brine	19 June 2023
ITALY	Trapani	STP	37°58′49.90′′ N	12°29′42.000′′ E	1000	NA	NA	NA	Salt	10 September 2022
	Motya	CP	37°51′47.02′′ N	12°29′4.781′′ E	1000	NA	NA	NA	Salt	23 September 2022
	Motya	SSR	37°51′46.70′′ N	12°29′1.263′′ E	1000	NA	NA	NA	Salt	23 September 2022
	Motya	SM16	37°51′46.90′′ N	12°28′59.985′′ E	1000	NA	NA	NA	Salt	23 September 2022
	Motya	SM19	37°51′46.90′′ N	12°28′59.98′′ E	1000	NA	NA	NA	Salt	23 September 2022

* Salinity was measured directly at the sampling site using a conventional refractometer; ** NA, not applicable.

**Table 2 microorganisms-12-01975-t002:** Properties and physicochemical parameters of brines from the various hypersaline saltern ponds in Vietnam.

Sampling Site Properties	Seawater	VS1new	VS7	VS8	HL345
Density, kg/m^3^	1.027	1.209	1.094	1.191	1.197
Total salinity, g/L	34.5	313.7	155.7	297.8	304.1
Ions, mmol/L					
Cl^−^	563	5140	2557	5160	4826
SO_4_^2−^	35	369	191	388	454
Br^−^	1	29	14	52	28
Na^+^	478	1912	796.9	433.9	1667
K^+^	11	48.9	20.5	11.3	44
Mg^2+^	61	1979	1073	2751	2031
Ca^2+^	11	1.8	4.0	2.2	2.1
Na^+^/Cl^−^ Ratio	0.857	0.372	0.312	0.084	0.345

**Table 3 microorganisms-12-01975-t003:** Alpha diversity inferred from 16S rRNA gene amplicon sequences for 19 brine, sediment and halite samples assayed in this study.

Sample ID	Number of Sequences	Number of ASVs ^1^	Coverage	Simpson Diversity Index	Shannon Diversity Index
VS8	17,865	180	1	65.82	4.54
VS1new	27,622	243	0.99	132.22	5.13
VS3	41,356	198	0.99	103.19	4.93
VCR	56,284	228	1	86.72	4.77
VFS	50,992	144	1	63.37	4.47
VS9 sed	56,620	542	0.99	320.05	6.00
VS2	48,344	463	0.99	292.39	5.89
VS5	46,810	206	0.99	93.40	4.92
VS4	56,518	434	0.99	148.66	5.41
VS6	51,319	380	0.99	179.35	5.46
Hai Lý 345	10,870	128	0.98	68.79	4.43
VS7	31,136	308	1	228.86	5.56
Hai Lý 450	4028	82	0.99	26.59	3.60
VS1	31,334	415	0.99	268.90	5.78
CP	61,118	59	1	12.12	2.99
SM16	41,430	199	0.99	133.43	5.07
SM19	50,621	200	1	73.96	4.89
SSR	26,570	133	1	76.98	4.60
STP	37,020	358	1	180.16	5.42
Total	747,857	4900			

^1^ The 99% threshold was used to emulate ASVs.

**Table 4 microorganisms-12-01975-t004:** Positive correlations between targeted phyla and physicochemical factors.

Phylum	Environmental Factor	Mantel’s *r*	Mantel’s *p*	FDR
*Nanohaloarchaeota*	S	0.899	0.001	0.013
*Halobacterota*	S	0.663	0.002	0.013
*Bacillota*	pH	0.503	0.006	0.020
*Bipolaricaulota*	pH	0.852	0.009	0.020
*Bacillota*	Redox	0.446	0.011	0.020
*Bipolaricaulota*	Redox	0.913	0.011	0.020
*Gemmatimonadota*	Redox	0.615	0.011	0.020
*Patescibacteria*	Redox	0.364	0.024	0.039
*Thermotogota*	pH	0.610	0.036	0.047
*Gemmatimonadota*	pH	0.506	0.038	0.047
*Desulfobacterota*	pH	0.335	0.046	0.048
*Deinococcota*	Redox	0.275	0.048	0.048
*Spirochaetota*	pH	0.253	0.049	0.047

## Data Availability

The sequencing dataset obtained in this study is freely available through the European Nucleotide Archive (ENA)/NCBI under the accession number PRJNA1142429, BioSample accessions SAMN42940649-67. All 16S rDNA sequences of *Nanohaloarchaeota*, obtained in various polysaccharide-degrading enrichments, were deposited in GenBank under accession number PQ139286-93 (https://submit.ncbi.nlm.nih.gov/subs/?search=SUB14640106 [accessed on 14 August 2024]). All the R scripts used to analyze the sequences and reproduce the pictures are available on: https://github.com/GiISP/Vietnam (accessed on 14 August 2024). All other data are available from the corresponding author upon reasonable request.

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
