# Peer review of "Unique Features of Extremely Halophilic Microbiota Inhabiting Solar Saltworks Fields of Vietnam"

_microorganisms, 2024, doi:10.3390/microorganisms12101975_

Round 1

Reviewer 1 Report

Comments and Suggestions for Authors

Cono et al. studied the microbial communities of solar saltwork fields of Vietnam and Italy. The environmental parameters were analyzed and the 16S rRNA based microbial community analysis were performed. results are analyzed in detail. It is of value to understand the unique role of the saltwork fields ecosystem. I recommended it to be published after a major revision.

1.     The sampling was completed in both Vietnam and Italy. However, information about Italy is not fully expressed, such as the title and abstract. The author needs to modify this.

2.     The conclusion section is too long. The author needs to summarize and refine this section.

Author Response

Comments 1. The sampling was completed in both Vietnam and Italy. However, information about Italy is not fully expressed, such as the title and abstract. The author needs to modify this.

Reply: Thank you for this suggestion. As requested, we have added all relevant information related to the Italian sampling in the Materials and Methods section. Please see the corresponding parts of the revised manuscript. We have also specified the Italian sampling sites in the Abstract section as much as possible, given the word limit of 200 words in this section. Regarding the title, the main and interesting finding of our paper is the presence of unique groups of ultra-small prokaryotes at the Vietnamese sites, which are absent from the Italian salterns studied for comparative purposes. Therefore, referring to the Italian sampling sites in the title seems irrelevant to the main message of the paper.

Comments 2. The conclusion section is too long. The author needs to summarize and refine this section.

Reply: Thank you for this suggestion. As requested, we have significantly shortened the conclusion section, which is actually only 32 lines long instead of the original 49 lines.

Reviewer 2 Report

Comments and Suggestions for Authors

1. There are too many figures in the manuscript. I think that authors can move Fig. 3 or Fig. 4 to supplementary files. And, authors can put together some pictures.

2. The section "Results and Discussion", I think that authors should do more comparisons with other published reports. For example, there are only one reference paper has been cited in 3.1-3.2. For my opinion, authors should divide the results and discussion to clear the importance of your paper.

3. Line 380, 416, 472, Pay attention to the orders. Line 380,416, repetition.

4. The section "Conclusion", too many words, authors should re-organise it.

Author Response

Comments 1. There are too many figures in the manuscript. I think that authors can move Fig. 3 or Fig. 4 to supplementary files. And, authors can put together some pictures.

Reply: Thank you for this suggestion. As requested, we have moved Fig. 3 and Fig. 4 in the Supplementary Materials as Supplementary Figure S1 and Figure S2. As for combining some pictures, we are not sure that this will be of sense, since even the original Figures are already quite visually complex.

Comments 2. The section "Results and Discussion", I think that authors should do more comparisons with other published reports. For example, there are only one reference paper has been cited in 3.1-3.2. For my opinion, authors should divide the results and discussion to clear the importance of your paper.

Thank you for this suggestion. As requested, in the section 3.1 we have added three additional sequences:

Carpenter, A. B. Origin and chemical evolution of brines in sedimentary basins. In SPE Annual Technical Conference and Exhibition, 1978, SPE-7504.

Wallmann, K., Suess, E., Westbrook, G. H., Winckler, G., Cita, M. B. (). Salty brines on the Mediterranean Sea floor. Nature, 1997, 387, 31-32.

Wallmann, K., Aghib, F. S., Castradori, D., Cita, M. B., Suess, E., Greinert, J., Rickert, D. Sedimentation and formation of secondary minerals in the hypersaline Discovery Basin, eastern Mediterranean. Marine Geology, 2002, 186, 9-28.

Moreover, in the Introduction section we have added the references to similar studies performed in solar salterns of Korea and Spain, as well as the reference to a recent review dedicated to the prokaryotic diversity in the manmade salterns worldwide:

Park, S. J., Kang, C. H., & Rhee, S. K. (2006). Characterization of the microbial diversity in a Korean solar saltern by 16S rRNA gene analysis. J. Microbiol. Biotechnol., 2006, 16, 1640-1645.

Casamayor, E. O., Massana, R., Benlloch, S., Øvreås, L., Díez, B., Goddard, V. J., Gasol, J. M., Joint, I., Rodriguez-Valera, F., Pedrós‐Alió, C. Changes in archaeal, bacterial and eukaryal assemblages along a salinity gradient by comparison of genetic fingerprinting methods in a multipond solar saltern. Environ. Microbiol. 2002, 4, 338-348.

Konstantinidis, K. T., Viver, T., Conrad, R. E., Venter, S. N., Rossello-Mora, R. Solar salterns as model systems to study the units of bacterial diversity that matter for ecosystem functioning. Curr. Opin. Biotechnol. 2022, 73, 151-157.

We carefully reviewed the commentary to separate the results and the discussion, but given that the other three reviewers were quite satisfied with the current structure of the article, we can confidently leave the original structure, especially since, in our opinion, such a presentation of the article is quite logical.

Comments 3. Line 380, 416, 472, Pay attention to the orders. Line 380,416, repetition.

Reply: Thank you very much for pointing us to this errors. We have corrected the heading correspondingly:

Line 380: 3.4. Bacterial diversity of brine, sediments and salt (halite) samples

Line 416: 3.5. Archaeal diversity of brine, sediments and salt (halite) samples

The heading of subsection 3.6. has been changed to as:

3.6. Diversity of nano-sized microbiota.

Comments 4. The section "Conclusion", too many words, authors should re-organise it.

Reply: Thank you for this suggestion. As requested, we have significantly shortened the conclusion section, which is actually only 32 lines long instead of the original 49 lines.

Reviewer 3 Report

Comments and Suggestions for Authors

The manuscript entitled “Unique features of extremely halophilic microbiota inhabiting solar saltwork fields of Vietnam» is devoted to halophilic prokaryotic communities study associated with stressful environments (salinity, intense ultraviolet radiation, elevated temperatures and fast desiccation processes) in northern Vietnam and in Italian salterns using  metabarcoding approach. I think that this manuscript describes a number of interesting and important results. To my mind this manuscript is topical and corresponding to the aims and scopes of the “Microorganisms” journal. I am ready to recommend it for publication after correcting several comments.

1.      L60 Decipher the phrase Culture-independent studies

2.      One of the well-known specialists in such habitats is Dmitry Sorokin https://www.researchgate.net/profile/Dimitry-Sorokin it is strange that the authors did not refer to his work in the introduction.

3.      In the introduction, more attention should be paid to the comparison of the two sampling sites and their common features should be shown.

4.      The purpose of the work should be stated more clearly

5.      In the results, the authors pay attention to the polysaccharide-degrading process, it is also worth paying more attention to this process in the introduction.

6.     L 476-479 should be moved to the introduction,

7.      In section 3.4. it is worth paying more attention to the comparison of the obtained results with the results of other researchers.

8.      In conclusion, it is worth adding a few phrases about the authors' opinion on the diversity of ultramicrobacteria in connection with extreme conditions, as well as the possible future prospects of the obtained results.

Author Response

Comments 1. L60 Decipher the phrase Culture-independent studies

Reply: Following this comment, we have added the extension phrase ‘…to culture-independent approaches, which use molecular techniques like sequencing to directly analyze microbial DNA from environmental samples without the need for cultivation’.

Comments 2. One of the well-known specialists in such habitats is Dmitry Sorokin https://www.researchgate.net/profile/Dimitry-Sorokin it is strange that the authors did not refer to his work in the introduction.

Reply: The Italian team of authors has established an active long-term collaboration with Dr. D.Y. Sorokin and recently has published many articles together with this world-known expert in halo(natrono)philic microorganisms. Nonetheless, the main interest of Dr. D.Y. Sorokin aims to the finding novel prokaryotes thriving in natural hypersaline and alkaline lakes, which have rather different environmental settings compared to intensively reusing manmade solar salters. Nonetheless, following this suggestion, we have added the reference to similar studies performed in solar salterns of Korea, as well as the references to changes in archaeal, bacterial assemblages along a salinity gradient and a recent review dedicated to the prokaryotic diversity in the manmade salterns worldwide, respectively:

Park, S. J., Kang, C. H., & Rhee, S. K. (2006). Characterization of the microbial diversity in a Korean solar saltern by 16S rRNA gene analysis. J. Microbiol. Biotechnol., 16(10), 1640-1645.

Casamayor, E. O., Massana, R., Benlloch, S., Øvreås, L., Díez, B., Goddard, V. J., Gasol, J. M., Joint, I., Rodriguez-Valera, F., Pedrós‐Alió, C. Changes in archaeal, bacterial and eukaryal assemblages along a salinity gradient by comparison of genetic fingerprinting methods in a multipond solar saltern. Environ. Microbiol. 2002, 4, 338-348.

Konstantinidis, K. T., Viver, T., Conrad, R. E., Venter, S. N., Rossello-Mora, R. Solar salterns as model systems to study the units of bacterial diversity that matter for ecosystem functioning. Curr. Opin. Biotechnol. 2022, 73, 151-157.

Comments 3&4. In the introduction, more attention should be paid to the comparison of the two sampling sites and their common features should be shown. The purpose of the work should be stated more clearly.

Reply: Both these suggestions were taken into account and the Introduction section was significantly modified by adding this part of text:

…This was done primarily to potentially identify geographically specific halophiles that could be linked to a location. Freshly collected halite samples were compared with brine and sediment samples for possible detection of geographically specific halophiles, regardless of the physical state of the collected samples. Finally, we integrated metabarcoding studies performed using freshly harvested halite and commercial salt samples obtained from Italian saltwork fields located in Trapani and Motya (western Sicily, Italy). This was done to obtain molecular signatures that could be implemented to track the salt samples using some geographical connotation of the product, linked to the production area…

Comments 5. In the results, the authors pay attention to the polysaccharide-degrading process, it is also worth paying more attention to this process in the introduction.

Reply: According to this comment, the final part of the Introduction section was expanded by adding this text:

However, there are some evidences for the presence and even dominance of Woesearchaeota (now order Woesearchaeales in the phylum Nanoarchaeota) in halo-alkaline lakes and hypersaline sediments and Aenigmatarchaeota (family Haloaenigmarchaeaceae) in slightly acidic hypersaline environments. Recently, some Nanohaloarchaeotawere stably cultivated in laboratory as members of polysaccharide-degrading consortia. Based on this finding, a similar enrichment approach was carried out with the Vietnamese samples to test whether some ultra-small prokaryotes could be an active ecophysiological component of extreme halophilic hydrolytic communities in the HKsf ecosystem.

Comments 6. L 476-479 should be moved to the introduction,

Reply: We moved these sentences to the Introduction section and added also a reference to a recently published article of Gutiérrez-Preciado et al.

Gutiérrez-Preciado, A., Dede, B., Baker, B. A., Eme, L., Moreira, D., López-García, P. Extremely acidic proteomes and metabolic flexibility in bacteria and highly diversified archaea thriving in geothermal chaotropic brines. Nat. Ecol. Evol. 2024, 1-14. doi.org/10.1038/s41559-024-02505-6

Comments 7. In section 3.4. it is worth paying more attention to the comparison of the obtained results with the results of other researchers.

Reply: we have added 8 additional references to the modified version of the manuscript.

Comments 8. In conclusion, it is worth adding a few phrases about the authors' opinion on the diversity of ultramicrobacteria in connection with extreme conditions, as well as the possible future prospects of the obtained results.

Reply: Following this suggestion, we added ensuing phrase:

The discovery of previously overlooked groups of halophilic ultrasmall CPR and DPANN organisms expanded our knowledge of their econiche spectra and suggests their active participation in biogeochemical cycling and functioning in hypersaline ecosystems, which may also imply control over the eubacterial and haloarchaea populations that bloom there.

Reviewer 4 Report

Comments and Suggestions for Authors

Major comments

This is a comprehensive 16S rRNA gene amplicon sequencing study of prokeryotic communities of saltworks, mostly from Vietnam, which has a different salt extraction procedure, resulting in unique elemental composition of the salterns. The work uses appropriate methodology (with the exception of a minor statistical issue, see below)  and is quite an important contribution to the ecology of hypersaline enviroments. My only major issue is that despite having ASVs, which generally provide close to species level information, the work remains exclusively at genus level descrption. It would have been more informative and interesting to provide the closest ASV-related species names, in genera where the biogeographic distribution of some species has been described, such as Halorubrum.

Minor comments

83 in HKsf the process is multiply repeated - rephrase

85 then into shallow pits - what are the pits lined/coated with?

193 Sequences were deprived of barcodes and primers  - rephrase

231 was continuously monitor- fix

253 was more interesting. Being 4.3 times - should be - was more interesting, being 4.3 times

343 As me mentioned above - fix

349 Several important factors with significant p and r values, such as salinity, pH, temperature, and oxygen availability, were measured - these are uncorrected p-values, and there are many hypotheses tested here, so a column with FDR should be included in Table 4

Comments on the Quality of English Language

There are a few grammatical errors, here and there.

Author Response

Comments 1. My only major issue is that despite having ASVs, which generally provide close to species level information, the work remains exclusively at genus level description. It would have been more informative and interesting to provide the closest ASV-related species names, in genera where the biogeographic distribution of some species has been described, such as Halorubrum.

Reply:

Thanks to the reviewer for this rather interesting and important comment. This was exactly what we wanted to demonstrate - the biogeographic distribution of some species of the genus Halorubrum and other haloarchaeal genera. The primers we used in the present study typically produced 440-445 bp amplicons of the prokaryotic 16S rRNA gene. Unfortunately, even containing the V3-V4 hypervariable region, such fragments are not long enough to obtain an unambiguous depth of resolution at the species level. This is especially relevant for the class Halobacteria, in which: (i) many genera consist of species that are very similar to each other (>99.0-99.5% identity); (ii) some genera (e.g. Halomicrobium and Haloarcula) consist of species with dramatic intraspecific polymorphism of the 16S rRNA genes. Their divergence (4.8–5.6%) is within the range of divergence typically found between genera in the order Halobacteriales (around 5–10%). However, we checked the depth of resolution of the Halorubrum-related ASVs again. As can be seen from the attached table ‘HalorubrumForReviewer4’, it is simply not possible to assign ASVs to one species or another.

Minor comments

83 in HKsf the process is multiply repeated – rephrase

We rephrased it with: ‘in HKsf the process is repeated many times’

85 then into shallow pits - what are the pits lined/coated with?

We rephrased it as to ‘typically coated with HDPE liner’

193 Sequences were deprived of barcodes and primers – rephrase

We rephrased it as: ‘The sequences were stripped of barcodes and primers.’

231 was continuously monitor- fix

Corrected.

253 was more interesting. Being 4.3 times - should be - was more interesting, being 4.3 times

We modified this sentence: ‘was more interesting. In fact, being 4.3 times saltier than seawater’

343 As me mentioned above – fix

Corrected.

349 Several important factors with significant p and r values, such as salinity, pH, temperature, and oxygen availability, were measured - these are uncorrected p-values, and there are many hypotheses tested here, so a column with FDR should be included in Table 4.

As requested, the Table 4 was modified accordingly by adding the column with FDR values. Moreover, we also slightly modified the text pointing to the observed FDR values:

… as indicated by high r values (0.899 and 0.663, respectively) and highly significant p values (0.001 and 0.002, respectively), the low False Discovery Rate (FDR) (0.013) suggested high relability of these results.

… (p < 0.01), an FDR value of 0.020 indicated moderate correlations that are still significant.

Round 2

Reviewer 1 Report

Comments and Suggestions for Authors

The manuscript has been revised as required.

Author Response

Thanks for the positive validation of our study.

Reviewer 2 Report

Comments and Suggestions for Authors

I have no more questions.

Author Response

(The authors gave the same response as above.)
